# Improving Long Document Topic Segmentation Models With Enhanced Coherence Modeling

**Hai Yu, Chong Deng, Qinglin Zhang, Jiaqing Liu, Qian Chen, Wen Wang**
Speech Lab, Alibaba Group
{yuhai.yu, w.wang}@alibaba-inc.com

## Abstract

Topic segmentation is critical for obtaining structured documents and improving downstream tasks such as information retrieval. Due to its ability of automatically exploring clues of topic shift from abundant labeled data, recent supervised neural models have greatly promoted the development of long document topic segmentation, but leaving the deeper relationship between coherence and topic segmentation underexplored. Therefore, this paper enhances the ability of supervised models to capture coherence from both **logical structure** and **semantic similarity** perspectives to further improve the topic segmentation performance, proposing **Topic-aware Sentence Structure Prediction (TSSP)** and **Contrastive Semantic Similarity Learning (CSSL)**. Specifically, the TSSP task is proposed to force the model to comprehend structural information by learning the original relations between adjacent sentences in a disarrayed document, which is constructed by jointly disrupting the original document at topic and sentence levels. Moreover, we utilize inter- and intra-topic information to construct contrastive samples and design the CSSL objective to ensure that the sentences representations in the same topic have higher similarity, while those in different topics are less similar. Extensive experiments show that Longformer with our approach significantly outperforms state-of-the-art (SOTA) methods. Our approach improves $F_1$ of SOTA by 3.42 (73.74 $\rightarrow$ 77.16) and improves $P_k$ by 1.11 points (15.0 $\rightarrow$ 13.89) on WIKI-727K and achieves an average relative reduction of 4.3% on $P_k$ on WikiSection. The average relative $P_k$ drop of 8.38% on two out-of-domain datasets also demonstrates the robustness of our approach[1].

## 1 Introduction

Topic segmentation aims to automatically segment the text into non-overlapping topically coherent parts (Hearst, 1994). Topic segmentation makes documents easier to read and understand, and also plays a key role in many downstream tasks such as information extraction (Prince and Labadié, 2007; Shtekh et al., 2018) and document summarization (Xiao and Carenini, 2019; Liu et al., 2022). Topic segmentation methods can be categorized into linear segmentation (Hearst, 1997), which yields a linear sequence of topic segments, and hierarchical segmentation (Bayomi and Lawless, 2018; Hazem et al., 2020), which produces a hierarchical structure with top-level segments divided into subsegments. We focus on linear topic segmentation in this work, especially for long documents.

Based on the definition of topics, each sentence in a topic relates to the central idea of the topic, and topics should be discriminative. Hence, two adjacent sentences from the same topic are more similar than those from different topics. Exploring this idea, prior unsupervised models mainly infer topic boundaries through computing text similarity (Riedl and Biemann, 2012b; Glavaš et al., 2016) or exploring topic representation of text (Misra et al., 2009; Du et al., 2013). Different from the shallow features carefully designed and used by unsupervised methods, supervised neural models can model deeper semantic information and explore clues of topic shift from labeled data (Badjatiya et al., 2018; Koshorek et al., 2018). Supervised models have achieved large gains on topic segmentation through pre-training language models (PLMs) (e.g., BERT) and fine-tuning on large-scale supervised datasets (Kenton and Toutanova, 2019; Lukasik et al., 2020; Zhang et al., 2021; Inan et al., 2022). Recently, (Arnold et al., 2019; Xing et al., 2020; Somasundaran et al., 2020; Lo et al., 2021) improve topic segmentation performance by explicitly modeling text coherence. However, these approaches either neglect context modeling beyond adjacent sentences (Wang et al., 2017), or require additional label information (Arnold et al., 2019;

---

[1] Our code is publicly available at https://github.com/alibaba-damo-academy/SpokenNLP/

Barrow et al., 2020; Lo et al., 2021; Inan et al., 2022), or impede learning sentence-pair coherence without considering both coherent and incoherent pairs (Xing et al., 2020). Moreover, compared to short documents, topic segmentation becomes more critical for understanding long documents, and coherence modeling for long document topic segmentation is more crucial.

Coherence plays a key role in understanding both logical structures and text semantics. Consequently, to enhance coherence modeling in supervised topic segmentation methods, we propose two auxiliary coherence-related tasks, namely, **Topic-aware Sentence Structure Prediction (TSSP)** and **Contrastive Semantic Similarity Learning (CSSL)**. We create disordered incoherent documents, then the TSSP task utilizes these documents and enhances learning sentence-pair structure information. The CSSL task regulates sentence representations and ensures sentences in the same topic have higher semantic similarity while sentences in different topics are less similar. Experimental results demonstrate that both TSSP and CSSL improve topic segmentation performance and their combination achieves further gains. Moreover, performance gains on out-of-domain data from the proposed approaches demonstrate that they also significantly improve generalizability of the model.

Large Language Models such as ChatGPT [2] have achieved impressive performance on a wide variety of NLP tasks. We adopt the prompts proposed by Fan and Jiang (2023) and evaluate ChatGPT on the WIKI-50 dataset (Koshorek et al., 2018). We find ChatGPT performs considerably worse than fine-tuning BERT-sized PLMs on long document topic segmentation (as shown in Appendix A).

Our contributions can be summarized as follows.

- We investigate supervised topic segmentation on long documents and confirm the necessity of exploiting longer context information.
- We propose two novel auxiliary tasks TSSP and CSSL for coherence modeling from the perspectives of both logical structure and semantic similarity, thereby improving the performance of topic segmentation.
- Our proposed approaches set new state-of-the-art (SOTA) performance on topic segmentation benchmarks, including long documents. Ablation study shows that both new tasks effectively

improve topic segmentation performance and they also improve generalizability of the model.

## 2 Related Work

### 2.1 Topic Segmentation Models

Both unsupervised and supervised approaches have been proposed before to solve topic segmentation. Unsupervised methods typically design features based on the assumption that segments in the same topic are more coherent than those that belong to different topics, such as lexical cohesion (Hearst, 1997; Choi, 2000; Riedl and Biemann, 2012b), topic models (Misra et al., 2009; Riedl and Biemann, 2012a; Jameel and Lam, 2013; Du et al., 2013) and semantic embedding (Glavaš et al., 2016; Solbiati et al., 2021; Xing and Carenini, 2021). In contrast, supervised models can achieve more precise predictions by automatically mining clues of topic shift from large amounts of labeled data, either by classification on the pairs of sentences or chunks (Wang et al., 2017; Lukasik et al., 2020) or sequence labeling on the whole input sequence (Koshorek et al., 2018; Badjatiya et al., 2018; Xing et al., 2020; Zhang et al., 2021). However, the memory consumption and efficiency of neural models such as BERT (Kenton and Toutanova, 2019) can be limiting factors for modeling long documents as their length increases. Some approaches (Arnold et al., 2019; Lukasik et al., 2020; Lo et al., 2021; Somasundaran et al., 2020) use hierarchical modeling from tokens to sentences, while others (Somasundaran et al., 2020; Zhang et al., 2021) use sliding windows to reduce resource consumption. However, both directions of methods may not be adequate for capturing the full context of long documents, which is critical for accurate topic segmentation.

### 2.2 Coherence Modeling

The NLP community has developed models for comprehending text coherence and tasks to measure their effectiveness, such as predicting the coherence score of documents (Barzilay and Lapata, 2008), predicting the position where the removed sentence was originally located (Elsner and Charniak, 2011) and restoring out-of-order sentences (Logeswaran et al., 2018; Chowdhury et al., 2021). Some researchers have aimed to improve topic segmentation models by explicitly modeling text coherence. However, **all of prior works consider coherence modeling for topic segmenta-**

[2] https://chat.openai.com

**tion only from a single perspective**. For example, Wang et al. (2017) ranked sentence pairs based on their semantic coherence to segment documents within the Learning-to-Rank framework, but they did not consider contextual information beyond two sentences. CATS (Somasundaran et al., 2020) created corrupted text by randomly shuffling or replacing sentences to force the model to produce a higher coherence score for the correct document than for its corrupt counterpart. However the fluency of the constructed document is too low so that the semantic information is basically lost. Xing et al. (2020) proposed to add the Consecutive Sentence-pair Coherence (CSC) task by computing the cosine similarity as coherence score. But no more incoherent sentence pairs are considered in CSC, except for those located at segment boundaries. Other methods (Arnold et al., 2019; Barrow et al., 2020; Lo et al., 2021; Inan et al., 2022) have used topic labels to constrain sentence representations within the same topic, but they require additional topic label information. In contrast to these works, **our work is the first to consider topical coherence as both text semantic similarity and logical structure (flow) of sentences**.

## 3 Methodology

In this section, we first describe our baseline model for topic segmentation (Section 3.1), then introduce our proposed Topic-aware Sentence Structure Prediction (TSSP) module (Section 3.2) and Contrastive Semantic Similarity Learning (CSSL) module (Section 3.3). Figure 1 illustrates the overall architecture of our topic segmentation model.

### 3.1 Baseline Model for Topic Segmentation

Our supervised baseline model formulates topic segmentation as a sentence-level sequence labeling task (Zhang et al., 2021)). Given a document represented as a sequence of sentences $[s_1, s_2, s_3, ..., s_n]$ (where $n$ is the number of sentences), the model predicts binary labels $[y_1, y_2, ..., y_{n-1}]$ corresponding to each sentence except for the last sentence, where $y_i = 1, i \in \{1, \cdots, n-1\}$ means $s_i$ is the last sentence of a topic and 0 means not.

Following prior works (Somasundaran et al., 2020; Zhang et al., 2021), we prepend a special token BOS before each sentence and the updated sentence is shown in Eq. 1, where $t_{i,1}$ is BOS and $|s_i|$ is the number of tokens in $s_i$.

$$s_i' = [t_{i,1}, t_{i,2}, ..., t_{i,|s_i|+1}] \quad (1)$$

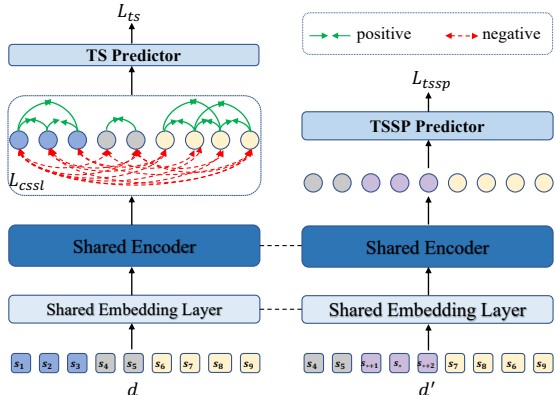

Figure 1: The overall architecture of our model. $s_i$ is $i$-th sentence in document $d$. $d'$ is the augmented data we construct corresponding to document $d$ (Section 3.2). TS denotes Topic Segmentation, TSSP denotes Topic-aware Sentence Structure Prediction (Section 3.2) and CSSL denotes Contrastive Semantic Similarity Learning (Section 3.3). $L_{ts}$, $L_{tssp}$ and $L_{cssl}$ denote the losses we describe in Section 3.

The token sequence for the document is embedded through the embedding layer and then fed into the encoder to obtain its contextual representations. We take the representation of each BOS $h_i$ as the sentence representation, as shown in Eq. 4. Then we apply a softmax binary classifier $g$ as in Eq. 3 on top of $h_i$ to compute the topic segmentation probability $p$ of each sentence. We use the standard binary cross-entropy loss function as in Eq. 2 to train the model.

$$L_{ts} = - \sum_{i=1}^{n-1} [y_i \ln p_i + (1-y_i) \ln(1-p_i)] \quad (2)$$

$$p_i = g(h_i) \quad (3)$$

$$h_i = Encoder(t_{i,1}) \quad (4)$$

### 3.2 Topic-aware Sentence Structure Prediction

Learning sentence representations that reflect inter-sentence coherence (inter-sentence relations) is critical for topic segmentation. Several tasks have been proposed in prior works for modeling sentence-pair relations. The Next Sentence Prediction (NSP) task in BERT (Kenton and Toutanova, 2019) predicts whether two segments appear consecutively in the same document or come from different documents, hence it fuses topic prediction and coherence prediction in one task. In order to better model inter-sentence coherence, the Binary Sentence Ordering (BSO) task in ALBERT (Lan et al., 2019) constructs input as two consecutive segments from the

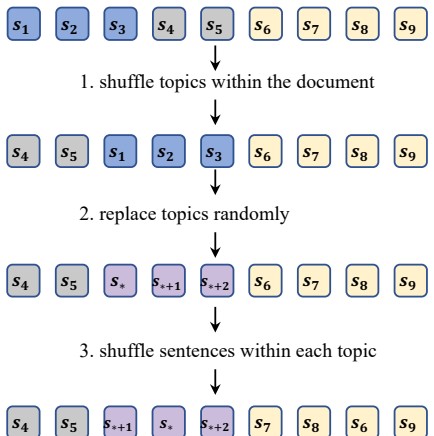

Figure 2: The process of constructing the augmented document (the bottom line) from the original document (the top line). $s_i$ denotes $i$-th sentence of the document. The sentences with the same colors are in the same topic. Sentences in light purple are topics from another document.

same document but 50% of the time their order is reversed. The ternary Sentence Structural Objective (SSO) in StructBERT (Wang et al., 2019) further increases the task difficulty of BSO by adding a class of sentence pairs from different documents.

All these tasks for learning inter-sentence coherence have not explored topic structures. Different from them, we propose a **Topic-aware Sentence Structure Prediction (TSSP)** task to help the model learn sentence representations with structural information that explore **topic structures** and hence are more suitable for topic segmentation.

**Data Augmentation** We tailor data augmentation techniques for topic segmentation. As depicted in the right half of Figure 1, we create an augmented document $d'$ from the original document $d$ and feed $d'$ into the shared encoder after the embedding layer to enhance inter-sentence coherence modeling. Different from the auxiliary coherence modeling approach proposed by Somasundaran et al. (2020) which merely forces the model to predict a lower coherence for the corrupted document than for the original document, we simultaneously perturb $d$ at both topic and sentence levels, constructing the augmented document to force the model to learn topic-aware inter-sentence structure information. Hence, our task is more challenging and the learned sentence representations are more suitable for topic segmentation. Figure 2 illustrates the process of constructing an augmented document. We first shuffle topics within the document, then randomly replace some topics with topics from other documents to increase diversity. Specifically, for a randomly selected subset of $p_1$ percent of the documents, we replace each topic in them with a topic snippet from other documents with a probability of $p_2$ and keep the same with the probability of $1 - p_2$. The default values of $p_1$ and $p_2$ are both 0.5. Finally, we shuffle the sentences in each topic to further increase the difficulty of the TSSP task.

**Sentence-pair Relations** After constructing the augmented document $d'$, we define the TSSP task as an auxiliary objective to assist the model to capture inter-sentence coherence, by learning the *original* structural relations between adjacent sentence pair $a$ and $b$ in the augmented *incoherent* document $d'$. We define three types of sentence-pair relations. The first type (label 0) is when $a$ and $b$ belong to different topics, indicating a topic shift. The second type (label 1) is when $a$ and $b$ are in the same topic and $b$ is the next sentence of $a$. The third type (label 2) is when $a$ and $b$ belong to the same topic but $b$ is not the next sentence of $a$. For example, the sequence of sentences in Figure 2 will be assigned with the TSSP labels as [1, 0, 2, 2, 0, 1, 2, 2]. We use $\tilde{y} = [\tilde{y}_0, \tilde{y}_1, \tilde{y}_2]$ to represent the one-hot encoding of the TSSP labels, where $\tilde{y}_j = 1, j \in \{0, 1, 2\}$ if the sentence pair belongs to the $j$-th category; otherwise, $\tilde{y}_j = 0$. For the TSSP task, we use the cross entropy loss function defined in Eq. 5, where $\tilde{y_{i,j}}$ denotes the label of the $i$-th sentence pair, and $\tilde{p_{i,j}}$ denotes the probability of the $i$-th sentence pair belonging to the $j$-th category.

$$L_{tssp} = -\sum_{i=1}^{n-1} \sum_{j=0}^{2} \tilde{y_{i,j}} \ln(\tilde{p_{i,j}}) \qquad (5)$$

### 3.3 Contrastive Semantic Similarity Learning

We assume that two sentences or segments from the same topic are inherently more coherent than those from different topics. Therefore, the motivation of our **Contrastive Semantic Similarity Learning (CSSL)** module is to adjust the sentence representation learning to grasp the **relative** high and low coherence relationship. Different from (Xing et al., 2020) which only calculates the cosine similarity of two adjacent sentences, our CSSL takes into account both *similar* and *dissimilar* sentence pairs to improve sentence representation learning.

**Construct Positive and Negative Samples** The upper left part of Figure 1 illustrates how CSSL

leverages contrastive learning to regulate sentence representations, which in turn influences the predictions of topic segmentation. To construct positive and negative sample pairs in the contrastive learning framework, prior works in NLP propose several data augmentation techniques such as word deletion and substitution (Wu et al., 2020), dropout (Gao et al., 2021), and adversarial attack (Yan et al., 2021). However, different from prior works that synthesize positive and negative pairs, we explore *natural* positive and negative pairs for topic segmentation, as two sentences or segments from the same topic are inherently more coherent than those from different topics. Accordingly, regarding each sentence in the document as the anchor sentence, we choose $k_1$ sentences in the same topic to constitute positive pairs and $k_2$ sentences from different topics as negative pairs based on the ordering of the distance of a sentence from the anchor sentence, starting from the nearest to the farthest. Recently, Gao et al. (2023) proposes a contrastive learning method for unsupervised topic segmentation. However, in their unsupervised method, similar and dissimilar sample pairs could be noisy due to lack of ground truth topic labels, which would not occur in our supervised settings.

**Loss Function** As illustrated in Figure 1, we utilize the following loss function to train our model and learn contrastive semantic representations of inter-topic and intra-topic sentences. $k_1$ and $k_2$ are hyperparameters that determine the number of sentences used to form positive and negative pairs, respectively. For each sentence representation $h_i$, $h_{i,j}^+$ denotes the $j$-th similar sentence in the same topic as sentence $i$, while $h_{i,j}^-$ denotes the $j$-th dissimilar sentence in a different topic from sentence $i$. We select sentences to form sentence pairs based on their distances to the anchor sentence, from closest to farthest. The objective of our loss function is to bring semantically similar neighbors closer and push away negative sentence pairs, as in Eq. 6. $\tau$ in Eq. 8 is a temperature hyper-parameter to scale the cosine similarity of two vectors, with a default value 0.1. In future work, in order to avoid pushing away sentence pairs in different topics but covering similar topical semantics, we plan to consider refining the loss, such as assigning loss weights based on their semantic similarity.

$$L_{cssl} = -\sum_{i=1}^{n} log(l_{cssl}^i) \quad (6)$$

$$l_{cssl}^i = \frac{\sum_{j=1}^{k_1} e^{sim(h_i, h_{i,j}^+)}}{\sum_{j=1}^{k_1} e^{sim(h_i, h_{i,j}^+)} + \sum_{j=1}^{k_2} e^{sim(h_i, h_{i,j}^-)}} \quad (7)$$

$$sim(x_1, x_2) = \frac{x_1^T x_2}{\|x_1\| \cdot \|x_2\|}/\tau \quad (8)$$

Combining Eq. 2, 5 and 6, we form the final loss function of our topic segmentation model as Eq. 9, where $\alpha_1$ and $\alpha_2$ are hyper-parameters used to adjust the loss weights.

$$L_{total} = L_{ts} + \alpha_1 L_{tssp} + \alpha_2 L_{cssl} \quad (9)$$

| Dataset | Docs | #Topics | #Sentences | #Tokens |
|---------|------|---------|------------|---------|
| WIKI-727K | 727,746 | 6.18 | 52.65 | 1356 |
| WikiSection | 23,129 | 6.98 | 57.69 | 1321 |
| WIKI-50 | 50 | 7.68 | 61.40 | 1544 |
| Elements | 118 | 7.71 | 23.81 | 1654 |

Table 1: Statistics of the Intra-domain and Out-of-domain datasets. #X denotes the average number of X per document.

## 4 Experiments

### 4.1 Experimental Setup

**Datasets** We conduct two sets of experiments to evaluate the effectiveness of our method, including intra-domain and out-of-domain settings. The details of the datasets are summarized in Table 1.

**Intra-domain Datasets** We use WIKI-727K (Koshorek et al., 2018) and English WikiSection (Arnold et al., 2019), which are widely used as benchmarks to evaluate the text segmentation performance of models. WIKI-727K is a large corpus with segmentation annotations, created by leveraging the manual structures of about 727K Wikipedia pages and automatically organizing them into sections. WikiSection consists of 38K English and German Wikipedia articles from the domains of *disease* and *city*, with the topic labeled for each section of the text. We use *en_city* and *en_disease* throughout the paper to denote the English subsets of disease and city domains, and use *WikiSection* to represent the collection of these two subsets. Each section is divided into sentences using the *PUNKT* tokenizer of the NLTK library[3]. Additionally, we utilize the newline information in WikiSection to only predict whether sentences with line breaks are

---
[3] https://www.nltk.org/

| Model | en_city | | | en_disease | | |
|---|---|---|---|---|---|---|
| | $F_1 \uparrow$ | $P_k \downarrow$ | $WD \downarrow$ | $F_1 \uparrow$ | $P_k \downarrow$ | $WD \downarrow$ |
| SEC>T+bloom (Arnold et al., 2019) | 71.6 | 14.4 | - | 56.6 | 26.8 | - |
| S-LSTM (Barrow et al., 2020) | 76.1 | 9.1 | - | 59.3 | 20.0 | - |
| BiLSTM+BERT (Xing et al., 2020) | - | 9.3 | - | - | 21.1 | - |
| Transformer$^2_{BERT}$ (Lo et al., 2021) | - | 9.1 | - | - | 18.8 | - |
| Tipster (Gong et al., 2022) | 79.8 | 8.3 | - | 62.2 | 14.2 | - |
| PEN-NS (Xia et al., 2022) | 80.0 | 8.0 | - | - | - | - |
| Naive LongT5-Base-DS (Inan et al., 2022) | - | 8.2 | - | - | 33.5 | - |
| Naive LongT5-Base-SS (Inan et al., 2022) | 73.1 | 9.2 | - | 38.8 | 24.8 | - |
| BERT-Base | 78.99 | 8.94 | 11.34 | 67.34 | 19.69 | 24.83 |
| +TSSP+CSSL (**ours**) | **80.16** | **8.22** | **10.19** | **68.26** | **18.29** | **22.06** |
| BigBird-Base | 80.49 | 8.21 | 10.24 | 70.61 | 16.73 | **20.44** |
| +TSSP+CSSL (**ours**) | **81.89** | **7.71** | **9.70** | **72.14** | **16.62** | 20.55 |
| LongformerSim | 79.75 | 9.85 | 11.75 | 66.66 | 19.60 | 22.74 |
| Longformer-Base | $82.19_{0.20}$ | $7.76_{0.13}$ | $9.74_{0.08}$ | $72.29_{0.31}$ | $16.66_{0.44}$ | $20.62_{0.79}$ |
| +CATS (Somasundaran et al., 2020)[†] | 82.30 | 7.84 | 9.80 | 72.00 | 16.21 | 19.95 |
| +CSC (Xing et al., 2020)[†] | 82.40 | 7.74 | 9.69 | 72.84 | 16.18 | 19.74 |
| +TSSP (**ours**) | 83.12 | 7.39 | 9.31 | 73.74 | 16.03 | 19.71 |
| +CSSL (**ours**) | 82.67 | 7.57 | 9.54 | 73.07 | 15.88 | 19.25 |
| +TSSP+CSSL (**ours**) | $\mathbf{83.19^*_{0.03}}$ | $\mathbf{7.38^*_{0.05}}$ | $\mathbf{9.29^*_{0.06}}$ | $\mathbf{74.17^*_{0.14}}$ | $\mathbf{15.44^*_{0.45}}$ | $\mathbf{19.05^*_{0.58}}$ |
| Pre-trained LongT5-Base-DS (Inan et al., 2022) | - | 6.8 | - | - | 15.3 | - |
| Pre-trained LongT5-Base-SS (Inan et al., 2022) | 82.3 | 7.1 | - | 68.3 | 15.0 | - |
| Pre-trained Longformer-Base | 84.70 | 6.83 | 8.52 | 76.02 | 14.15 | 17.28 |
| +TSSP+CSSL (**ours**) | **85.14** | **6.48** | **8.26** | **77.33** | **13.66** | **16.70** |

Table 2: Performance of baselines and w/ our methods on *en_city* and *en_disease* test sets of WikiSection. LongformerSim denotes Longformer-Base that uses cosine similarity of neighbor sentences as the predictor. Pre-trained Longformer-Base denotes further pre-training Longformer-Base with WIKI-727K training set and then fine-tuning with WikiSection training set. † denotes training Longformer-Base with the corresponding auxiliary task described in Section 2.2. Max sequence length for BigBird-Base and Longformer-Base is 2048. $x$ and $y$ in $x_y$ denote mean and standard deviation from three runs with different random seeds. ∗ indicates the gains from +TSSP+CSSL over Longformer-Base are statistically significant with $p < 0.05$.

topic boundaries, which is beneficial to alleviate the class imbalance problem.

**Out-of-domain Datasets** Following prior work (Xing et al., 2020), to evaluate the domain transfer capability of our model, we fine-tune the model on the training set of *WiKiSection* dataset (union of *en_city* and *en_disease*) due to its distinct domain characteristics. Then we evaluate its performance on two other datasets, including WIKI-50 (Koshorek et al., 2018) and Elements (Chen et al., 2009), which have different domain distributions from WikiSection. Specifically, WIKI-50 consists of 50 samples randomly selected from Wikipedia, while Elements consists of 118 samples which are also extracted from Wikipedia but focuses on chemical elements.

**Evaluation Metrics** Following prior works, we use three standard evaluation metrics for topic segmentation, namely, positive F$_1$, $P_k$ (Beeferman et al., 1999), and WindowDiff (*WD*) (Pevzner and Hearst, 2002) [4]. To simplify notations, we use F$_1$ and *WD* throughout the paper to denote posi-

---
[4]We use https://segeval.readthedocs.io/ to compute $P_k$ and *WD*

tive F$_1$ and WindowDiff. F$_1$ is calculated based on precision and recall of correctly predicted topic segmentation boundaries. The $P_k$ metric is introduced to address some limitations of positive F$_1$, such as the inherent trade-off between precision and recall as well as its insensitivity to near-misses. *WD* is proposed by Pevzner and Hearst (2002) as a supplement to $P_k$ to avoid being sensitive to variations in segment size distribution and over-penalizing near-misses. By default, the window size for both $P_k$ and *WD* is equal to half the average length of actual segments. Lower $P_k$ and *WD* scores indicate better algorithm performance.

**Baseline Models** Although Transformer (Vaswani et al., 2017) has become the SOTA architecture for sequence modeling on a wide variety of NLP tasks and transformer-based PLMs such as BERT (Devlin et al., 2019) become dominant in NLP, the core self-attention mechanism has quadratic time and memory complexity to the input sequence length (Vaswani et al., 2017), limiting the max sequence length during pre-training (e.g., 512 for BERT) for a balance between performance and memory usage. As shown in Table 1, the avg. num-

ber of tokens per document of each dataset exceeds 512 and hence these datasets contain long documents. We tailor the backbone model selection for long documents. Prior models using BERT-like PLMs for topic segmentation either truncate long documents into the max sequence length or use a sliding window. These approaches may degrade performance due to losing contextual information. Consequently, we first evaluate BERT-Base (Devlin et al., 2019) and several competitive efficient transformers on the WikiSection dataset, including BigBird-Base (Zaheer et al., 2020) and Longformer-Base (Beltagy et al., 2020). As shown in Table 2, Longformer-Base achieves **82.19** and **72.29** $F_1$, greatly outperforming BERT-Base (78.99 and 67.34 $F_1$) by **(+3.2, +4.95)** $F_1$ and BigBird-Base (80.49 and 70.61 $F_1$). Hence we select Longformer-Base as the encoder for the main experiments. To compare with our coherence-related auxiliary tasks, we evaluate Longformer-Base on WikiSection with the prior auxiliary CATS or CSC task in Section 2.2. In addition, following Inan et al. (2022), we evaluate the pre-trained settings where we first pre-train Longformer on WIKI-727K and then fine-tune on WikiSection. Under the domain transfer setting, we cite the results in (Xing et al., 2020). Note that all the baselines we include for comparisons are well-established and exhibit top performances on these benchmarks.

**Implementation Details** To investigate the efficacy of exploring longer context for topic segmentation, we conducted additional evaluations on WikiSection using maximum sequence lengths of 512, 1024, and 4096, alongside the default 2048. For documents longer than the max sequence length, we use a sliding window to take the last sentence of the prior sample as the start sentence of the next sample. We run the baseline Longformer-Base and w/ our model (i.e., Longformer-Base+TSSP+CSSL) three times with different random seeds and report means and standard deviations of the metrics. Details of hyperparameters are in Appendix B.

## 4.2 Main Results

**Intra-domain Performance** Table 2 and Table 3 show the performance of baselines and w/ our approaches on WikiSection and WIKI-727K test sets, respectively. The results of Longformer-Base and LongformerSim in Table 2 show that using cosine similarity alone is insufficient to predict the topic segmentation boundary. Longformer-

| Model | WIKI-727K | | |
| --- | --- | --- | --- |
| | $F_1 \uparrow$ | $P_k \downarrow$ | $WD \downarrow$ |
| Bi-LSTM (Koshorek et al., 2018) | - | 22.13 | - |
| Cross-segment BERT (Lukasik et al., 2020) | 66.0 | - | - |
| Hier. BERT (Lukasik et al., 2020) | 66.5 | - | - |
| CATS (Somasundaran et al., 2020) | - | 15.95 | - |
| Seq-BERT-Base (Zhang et al., 2021)[†] | 70.39 | 17.35 | 18.50 |
| Seq-ELECTRA-Base (Zhang et al., 2021)[†] | 73.74 | 15.83 | 17.01 |
| Naive LongT5-Base-DS (Inan et al., 2022) | - | 15.4 | - |
| Naive LongT5-Base-SS (Inan et al., 2022) | - | 15.0 | - |
| Longformer-Base | $76.27_{0.07}$ | $14.40_{0.03}$ | $15.50_{0.03}$ |
| +TSSP (**ours**) | 76.57 | 14.13 | 15.20 |
| +CSSL (**ours**) | 76.30 | 14.28 | 15.40 |
| +TSSP+CSSL (**ours**) | $\mathbf{77.16}^{*}_{0.06}$ | $\mathbf{13.89}^{*}_{0.02}$ | $\mathbf{14.99}^{*}_{0.02}$ |

Table 3: Performance of baselines and w/ our methods on the WIKI-727K test set. † represents our reproduced results. $x$ and $y$ in $x_y$ denote mean and standard deviation from three runs with different random seeds. ∗ indicates the gains from +TSSP+CSSL over Longformer-Base are statistically significant with $p < 0.05$.

Base already outperforms all baselines in the first group of Table 2 and Table 3 and BERT-Base and BigBird-Base. Training BERT-Base, BigBird-Base and Longformer-Base with our TSSP or CSSL task achieves further gains. Table 2 shows that our TSSP and CSSL both outperform CATS or CSC auxiliary tasks. More importantly, combining TSSP and CSSL exhibits complementary effects, as it achieves further gains and sets the new SOTA, confirming the necessity of modeling both sentence structure and text semantic similarity for modeling text coherence and in turn for topic segmentation. On WikiSection, +TSSP+CSSL improves BERT-Base by **(+1.17, +0.92)** $F_1$, BigBird-Base by **(+1.4, +1.53)** $F_1$, and Longformer-Base by **(+1.0, +1.88)** $F_1$. On WIKI-727K, +TSSP+CSSL improves Longformer-Base by **+0.89** $F_1$. In addition, utilizing pre-training data also improves the performance on WikiSection, by **(+1.95, +3.16)** $F_1$. Finally, our new SOTA reduces $P_k$ of old SOTA by **1.11** points on WIKI-727K (15.0→ 13.89) and achieves an average relative reduction of 4.3% on $P_k$ on WikiSection. It is also important to note that our proposed TSSP and CSSL are agnostic to document lengths and are also applicable to models and datasets for short documents, which is verified by their gains on both short and long document subsets of WIKI-727K test set (as shown in Appendix C).

**Domain Transfer Performance** Table 4 shows the performance of the baselines and w/ our method on the out-of-domain WIKI-50 and Elements test sets. Longformer-Base already achieves **5.51** point reduction on $P_k$ on Elements over the prior best performance from supervised models, and our approach further improves $P_k$ by **2.83** points. While

| Model | WIKI-50 | | | Elements | | |
|---|---|---|---|---|---|---|
| | $F_1 \uparrow$ | $P_k \downarrow$ | $WD \downarrow$ | $F_1 \uparrow$ | $P_k \downarrow$ | $WD \downarrow$ |
| BayesSeg (Eisenstein and Barzilay, 2008) | - | 49.2 | - | - | 35.6 | - |
| GraphSeg (Glavaš et al., 2016) | - | 63.6 | - | - | 49.1 | - |
| Sector (Arnold et al., 2019) | - | 28.6 | - | - | 42.8 | - |
| CATS (Somasundaran et al., 2020) | - | 29.3 | - | - | 45.2 | - |
| BiLSTM+BERT (Xing et al., 2020) | - | 26.8 | - | - | 39.4 | - |
| Longformer-Base | $46.45_{0.85}$ | $28.76_{0.28}$ | $31.16_{0.63}$ | $55.13_{2.16}$ | $33.89_{3.01}$ | $44.99_{2.81}$ |
| +CATS (Somasundaran et al., 2020)[†] | 43.96 | 28.11 | 29.82 | 57.48 | 31.45 | **41.75** |
| +CSC (Xing et al., 2020)[†] | 48.18 | 28.88 | 31.20 | 55.18 | 33.26 | 42.51 |
| +TSSP+CSSL (**ours**) | $\mathbf{52.29}^{*}_{1.33}$ | $\mathbf{25.70}^{*}_{0.65}$ | $\mathbf{27.69}^{*}_{0.60}$ | $\mathbf{58.54}_{2.51}$ | $\mathbf{31.06}_{3.05}$ | $43.91_{1.40}$ |

Table 4: Performance of baselines and w/ our methods under domain transfer setting. BERT and Longformer are Base size. The training set of WikiSection is used for fine-tuning. † denotes fine-tuning Longformer with the corresponding auxiliary task described in Section 2.2. $x$ and $y$ in $x_y$ denote mean and standard deviation from three runs with different random seeds. ∗ indicates the gains from +TSSP+CSSL over Longformer-Base are statistically significant with $p < 0.05$.

Longformer-Base does not perform best on WIKI-50, incorporating our method achieves **+5.84** $F_1$ gain and **3.06** point gain on $P_k$, setting new SOTA on WIKI-50 and Elements for both unsupervised and supervised methods. Overall, the results demonstrate that our proposed method not only greatly improves the performance of a model under the intra-domain setting, but also remarkably improves the generalizability of the model.

**Inference Speed** Our proposed TSSP and CSSL do not bring any additional computational cost to inference and do not change the inference speed. We randomly sample 1K documents from WIKI-727K test set and measure the inference speed on a single Tesla V100 GPU with $batch\_size = 1$. On average, BERT-Base with max sequence length 512 processes 19.5K tokens/sec while Longformer-Base with max sequence length 2048 processes 15.9K tokens/sec. This observation is consistent with the findings in (Tay et al., 2020) that Longformer does not show a speed advantage over BERT until the input length exceeds 3K tokens.

## 5 Analysis

**Effect of Context Size** To study the effect of the context size, we evaluate Longformer with max sequence length of 512, 1024, 2048 and 4096 on WikiSection. We evaluate the effectiveness of our proposed methods using the corresponding max sequence length to investigate whether they remain effective with different context sizes. As can be seen from Figure 3, the topic segmentation $F_1$ gradually improves as the context length increases. Among them, the effect of increasing the input length from 512 to 1024 is the largest with $F_1$ on *en_city* and *en_disease* improved by **+2.54** and

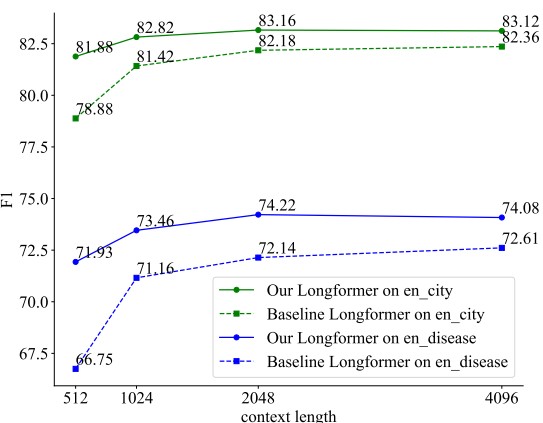

Figure 3: F1 of baseline Longformer and our Longformer (i.e., Longformer+TSSP+CSSL) on the WikiSection *en_city* and *en_disease* test sets as the context length increases.

**+4.41** respectively. Considering that the average document length of WikiSection is 1321, we infer that capturing more context information is more beneficial to topic segmentation on long documents. We also observe that compared to Longformer-Base, Longformer+TSSP+CSSL yields consistent improvements across different input lengths on both *en_city* and *en_disease* test sets. These results suggest that our methods are effective at enhancing topic segmentation across various context sizes and can be applied to a wide range of data sets. **Ablation Study of TSSP and CSSL** Figure 4 shows ablation study of TSSP and CSSL on the WikiSection dev set, respectively. Figure 4(a) demonstrates effectiveness of the three classification tasks in the TSSP task (Section 2.1). Compared to SSO and CATS, TSSP helps the model learn better inter-sentence relations and both intra- and inter-topic

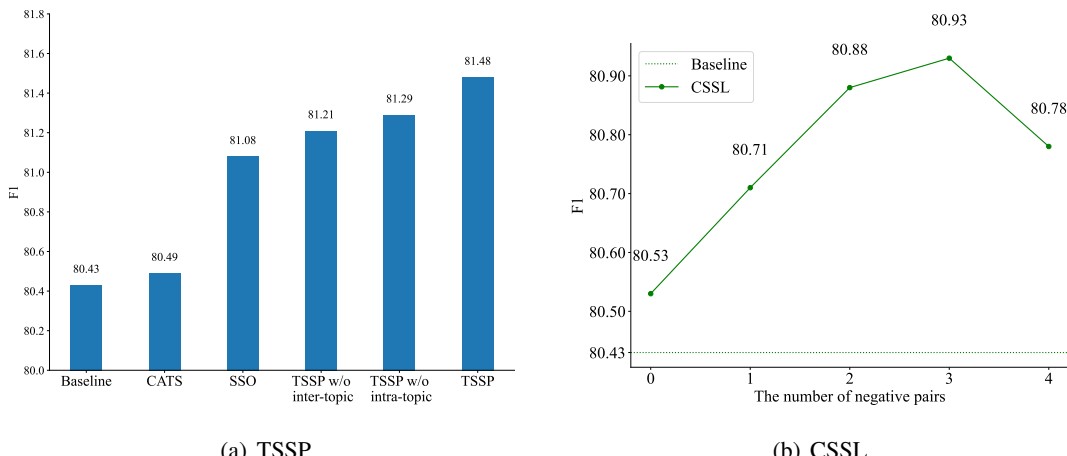

|     |     |
|:---:|:---:|
| (a) TSSP | (b) CSSL |

Figure 4: Ablation study of Longformer with our method on the WikiSection dev set with (a) TSSP and (b) CSSL separately. CATS and SSO in Figure (a) are previously proposed auxiliary tasks where CATS denotes Coherence-Aware Text Segmentation (Section 2.2) and SSO denotes Sentence Structural Objective (Section 3.2). In Figure (a), TSSP w/o inter-topic means without category 0 and w/o intra-topic means category 1 and 2 in Section 3.2. In Figure (b), 0 negative pairs represents fine-tuning with just the CSC task (Section 2.2).

structure labels are needed to improve performance. Figure 4(b) illustrates the impact of varying numbers of negative sample pairs for CSSL on Longformer. We find that adding similarity-related auxiliary tasks improves the performance. Compared with CSC, CSSL focuses on sentences of the same and different topics when learning sentence representations. As the number of negative samples increases, the model performance improves and optimizes at $k_2=3$. Gain from TSSP is slightly larger than that from CSSL, indicating that comprehending structural information contributes more to coherence modeling. We speculate that encoding the entire topic segment into a semantic space to learn contrastive representation may help detecting topic boundaries, which we plan to explore in future work.

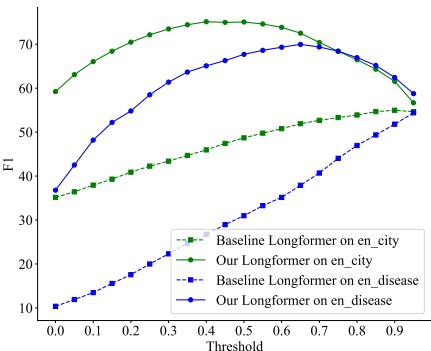

Figure 5: $F_1$ from using cosine similarity of two adjacent sentence representations to predict topic boundaries, with different thresholds on WikiSection dev set.

**Similarity of Sentence-pair Representations** To investigate impact of coherence-related auxiliary tasks on sentence representation learning, we calculate cosine similarity of adjacent sentence representations for predicting topic boundaries. We compute $F_1$ of baselines and our Longformer (i.e., Longformer-Base+TSSP+CSSL) on *en_city* and *en_disease* dev sets. As shown in Figure 5, compared to Longformer-Base, our model achieves higher $F_1$, indicating that sentence representations learned with our methods are more relevant to topic segmentation and are better at distinguishing sentences from different topics. We also explore combining probability and similarity to predict topic boundaries in Appendix D but find no further gain. The results suggest that the model trained with TSSP+CSSL covers more features than similarity.

## 6 Conclusion

Comprehending text coherence is crucial for topic segmentation, especially on long documents. We propose the Topic-aware Sentence Structure Prediction and Contrastive Semantic Similarity Learning auxiliary tasks to enhance coherence modeling. Experimental results show that Longformer trained with our methods significantly outperforms SOTAs on two English long document benchmarks. Our methods also significantly improve generalizability of the model. Future work includes extending our approach to spoken document topic segmentation and other segmentation tasks at various levels of granularity and with modalities beyond text.

## Limitations

Although our approach has achieved SOTA results on long document topic segmentation, further research is required on how to more efficiently model even longer context. In addition, our method needs to construct augmented data for the TSSP task, which will take twice the training time.

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

## A ChatGPT for Topic Segmentation in Long Document

To investigate the performance of ChatGPT on long document topic segmentation, we adopt the prompts proposed by Fan and Jiang (2023) and evaluate ChatGPT on the WIKI-50 test set. The prompts are shown in Table 5. We set temperature as 0 to ensure the consistency of ChatGPT. The post-processing strategy for ChatGPT remains unchanged to obtain the formatted output. Different from Fan and Jiang (2023), we change the key word **dialogue** to **document** in the prompt and try to use one-shot prompt to see if it can improve performance.

Table 6 shows the results of ChatGPT with different prompts and Longformer with supervised data. Firstly, the results show that the generative prompt can achieve higher performance than the discriminative prompt, which is consistent with the conclusion of Fan and Jiang (2023) that representing structure directly can better leverage the generation ability of ChatGPT. Additionally, incorporating a single example in the generative prompt can further improve 2-point $F_1$, which indicates one-shot prompt can better stimulate the in-context learning ability of Large Language Models. However, while ChatGPT-GP$_{one}$ achieves an $F_1$ metric that is 4.6 points higher than Longformer$_{ood}$, its' $P_k$ and $WD$ metric are significantly worse due to the high false recall rate of topic boundaries. This suggests that how to fully apply the ability of Large Language Models to the topic segmentation in long documents remains to be further explored. Finally, compared with ChatGPT, the significant improvement of Longformer$_{iid}$ shows the key role supervised data plays in the topic segmentation task.

## B Training Details

Our experiments are implemented with *transformers* package[5]. The model parameters are initialized with corresponding pre-trained parameters. The initial learning rate is $5e-5$ and the dropout probability is 0.1. AdamW (Loshchilov and Hutter, 2017) is used for optimization. The batch size for WIKI-727K and WikiSection is 4 and 8, and the epoch for WIKI-727K and WikiSection is 3 and 5 respectively. We set the number of positive pair in CSSL as $k_1 = 1$ and carry out grid-search of loss weight $\alpha_1, \alpha_2 \in [0.5, 1.0]$, $k_2 \in [1, 2, 3, 4]$ on dev set. The final configuration in the two benchmarks is $k_2 = 3, \alpha_1 = 0.5$. $\alpha_2$ performs best on WikiSection when set to 1.0 while on WIKI-727K $\alpha_2 = 0.5$ performs best.

---

[5] https://github.com/huggingface/transformers

| Type | Prompts for Document Topic Segmentation |
|---|---|
| Discriminative | The following is a document. Give each utterance a binary label, where 1 indicates that the utterance starts a new topic. please output the result of the sequence annotation as a python list.
0: $s_1$
1: $s_2$
...
$n-1$: $s_n$ |
| Generative | Please identify several topic boundaries for the following document and each topic consists of several consecutive utterances. please output in the form of {topic i:[], ... ,topic j:[]}, where the elements in the list are the index of the consecutive utterances within the topic, and output even if there is only one topic.
0: $s_1$
1: $s_2$
...
$n-1$: $s_n$ |

Table 5: Prompts for Document Topic Segmentation. $n$ denotes the number of sentences and $s_i$ denotes $i$-th sentence in the document.

| Model | WIKI-50 | | |
|---|---|---|---|
| | $F_1 \uparrow$ | $P_k \downarrow$ | $WD \downarrow$ |
| ChatGPT-DP$_{zero}$ | 23.28 | 52.83 | 66.63 |
| ChatGPT-GP$_{zero}$ | 49.59 | 37.79 | 46.18 |
| ChatGPT-DP$_{one}$ | 21.91 | 54.86 | 71.40 |
| ChatGPT-GP$_{one}$ | **51.61** | **37.19** | **45.21** |
| Longformer$_{ood}$ | 47.01 | 29.04 | 31.41 |
| Longformer$_{iid}$ | **76.32** | **11.01** | **11.82** |

Table 6: Comparison of ChatGPT and Longformer on the WIKI-50 test set. Longformer$_{ood}$ denotes fine-tuning Longformer on WikiSection and Longformer$_{iid}$ denotes fine-tuning Longformer on WIKI-727K. DP and GP are short for discriminative prompt and generative prompt, respectively. *zero* and *one* denote zero-shot and one-shot prompting settings.

## C Performance of the Proposed Approach on Short and Long Documents

It is important to note that our proposed TSSP and CSSL approaches are agnostic to document lengths and are applicable to models and datasets for short documents. In order to evaluate the performance of our proposed approach on various document lengths, we partition the WIKI-727K test set into a short document subset (18310 samples) and a long document subset (54922 samples) according to whether the number of tokens in a document is less than 512 or not. The results from the baseline Longformer and Longformer with our approaches

| Model | WIKI-727K$_{short}$ | | | WIKI-727K$_{long}$ | | |
|---|---|---|---|---|---|---|
| | $F_1 \uparrow$ | $P_k \downarrow$ | $WD \downarrow$ | $F_1 \uparrow$ | $P_k \downarrow$ | $WD \downarrow$ |
| Longformer-Base | 83.36$_{0.09}$ | 11.58$_{0.10}$ | 11.78$_{0.11}$ | 75.20$_{0.06}$ | 15.34$_{0.02}$ | 16.76$_{0.03}$ |
| +TSSP+CSSL(**ours**) | **83.97**$_{0.09}$ | **11.13**$^*_{0.06}$ | **11.30**$^*_{0.07}$ | **76.18**$^*_{0.05}$ | **14.81**$^*_{0.01}$ | **16.22**$^*_{0.01}$ |

Table 7: The performance of Longformer-Base and our approach on short and long document subsets of WIKI-727K test set.∗ indicates the gains from +TSSP+CSSL over Longformer-Base are statistically significant with $p < 0.05$.

| Model | Score | en_city | | | en_disease | | |
|---|---|---|---|---|---|---|---|
| | | $F_1 \uparrow$ | $P_k \downarrow$ | $WD \downarrow$ | $F_1 \uparrow$ | $P_k \downarrow$ | $WD \downarrow$ |
| Longformer | *Prob Only* | 82.18 | 7.87 | 9.82 | 72.33 | 17.23 | 22.30 |
| | *Prob and Sim* | 82.05 | 7.97 | 10.07 | 72.69 | 16.39 | 20.71 |
| +TSSP+CSSL | *Prob Only* | **83.16** | **7.33** | **9.24** | 74.19 | **16.29** | **20.29** |
| | *Prob and Sim* | 83.08 | 7.52 | 9.47 | **74.31** | 16.44 | 20.76 |

Table 8: The results of combing the probability and cosine similarity to predict topic boundary on *en_city* and *en_disease*. *Prob Only* denotes only using probability. *Prob and Sim* denotes compute score as Formula 10.

(+TSSP+CSSL) are shown in Table 7. Our approach significantly improves the baseline on both short and long documents. Notably, the gains from our approach are larger on long documents, suggesting that our coherence modeling benefits long document topic segmentation even more. This is consistent with our hypothesis.

## D Ensemble Probability and Similarity

As shown in Formula 10, we try to combine the cosine similarity of neighbor sentence representations (*Sim*) and model probability (*Prob*) to get the final score to infer topic boundary. Specifically, we get

different thresholds at intervals of 0.05 from 0 to 1. Then we choose the threshold when $F_1$ reaches the best in the dev set, and finally use this threshold to obtain the effect on the test set. The results shown in Table 8 suggest that the score combining similarity of neighbor sentence representations and model probability lower the performance slightly. We speculate that the boundary probability prediction and auxiliary coherence tasks are performed during training simultaneously, therefore the model has incorporated more features than just similarity.

$$Score = \frac{1}{2} * (Prob + Sigmoid(-1 * Sim)) \quad (10)$$

$$Sigmoid(x) = \frac{1}{1 + e^{-x}} \quad (11)$$