# OpenReview forum: "Improving Long Document Topic Segmentation Models With Enhanced Coherence Modeling"
_EMNLP/2023/Conference — EMNLP 2023 Main_

### Official Review · Reviewer_wwEk · 2023-08-05

**Soundness:** 4

**Excitement:**

3: Ambivalent: It has merits (e.g., it reports state-of-the-art results, the idea is nice), but there are key weaknesses (e.g., it describes incremental work), and it can significantly benefit from another round of revision. However, I won't object to accepting it if my co-reviewers champion it.

**Missing References:**

[1] Wu, Xiaobao, Anh Tuan Luu and Xinshuai Dong. “Mitigating Data Sparsity for Short Text Topic Modeling by Topic-Semantic Contrastive Learning.” Conference on Empirical Methods in Natural Language Processing (2022).
[2] Luo, Zheheng, Lei Liu, Qianqian Xie and Sophia Ananiadou. “Graph Contrastive Topic Model.” ArXiv abs/2307.02078 (2023): n. pag.

**Paper Topic And Main Contributions:**

This paper proposes an innovative method to enhance the process of topic segmentation in long documents, improving their structure and the effectiveness of information retrieval tasks. The approach focuses on enhancing semantic coherence in topic segmentation by introducing two new methods. The first, Topic-aware Sentence Structure Prediction (TSSP), is designed to improve the model's understanding of the sentence structure in relation to the topic. The second, Contrastive Semantic Similarity Learning (CSSL), focuses on learning the semantic similarity or difference between sentences. The experimental results suggests that the implementation of these methods could lead to significant improvements in the topic segmentation process.

**Questions For The Authors:**

1. The paper does not seem to reference recent advancements in topic contrastive learning, as outlined in [1][2]. Could you explain why these were not considered in your study? Would incorporating these advancements impact your methods and results?
2. Could you discuss the potential impact of your research on the broader community? How do these methods contribute to the advancement of the field?

**Reasons To Accept:**

1. Writing: The paper is well-structured and clearly presents the problem statement, its importance, and the proposed solutions.
2. Idea: The paper proposes innovative methods (TSSP and CSSL) to enhance topic segmentation in long documents. These methods address existing limitations in supervised neural models, particularly in exploring the deeper relationship between semantic coherence and topic segmentation.
3. Experiment: Experimental results demonstrate the improving of the performance of topic segmentation in long documents.

**Reasons To Reject:**

- Limited Impact: The proposed methods are limited in their application to topic segmentation, which have little impact on the broader research or application community.
- Missing references: The authors ignore the recent advancements in topic contrastive learning, such as [1][2].

**Reproducibility:**

4: Could mostly reproduce the results, but there may be some variation because of sample variance or minor variations in their interpretation of the protocol or method.

**Reviewer Confidence:**

3: Pretty sure, but there's a chance I missed something. Although I have a good feel for this area in general, I did not carefully check the paper's details, e.g., the math, experimental design, or novelty.

**Typos Grammar Style And Presentation Improvements:**

Table 2,3,4 are too small to read comfortably; consider enlarging the table, increasing the font size, and improving the contrast for better readability.

---

> ### Author Rebuttal · Authors · 2023-08-29
>
> We thank the reviewer for the thoughtful reviews and valuable feedback. Below we address all of your questions and concerns.
>
> **Q1: What's the potential impact of this research on the broader community.**
>
> **Response**: First of all, topic segmentation is a critical task for structuring documents and topic segmentation plays a key role in many downstream tasks such information extraction [1] and document summarization [2] (Line 042-047 of our paper). Our methods significantly improve topic segmentation performance, hence our work will significantly benefit document understanding and many downstream tasks.
>
> Furthermore, it is worth noting that our proposed TSSP and CSSL methods, with adjustments, can be applied to a broad spectrum of segmentation tasks at various levels of granularity and with modalities other than text. For example, our methods can be applied to models for paragraph segmentation or section segmentation on text or scene segmentation on narrative texts [3]. Our methods can also be applied to the computer vision field such as scene boundary detection of video [4]. We believe that our work holds great potential on the broader community. These topics can be explored in future directions to demonstrate broader impacts of our work.
>
>
> **Q2: Missing the recent advancements in topic contrastive learning. Would incorporating these advancements impact the methods and results?**
>
> **Response**: Thank you for recommending paper [5] and paper [6]. Paper [5] was published on arxiv on July 5, 2023, which is after EMNLP 2023 direct paper submission deadline June 23, 2023. Consequently, we did not cite this work. Paper [6] focuses on producing high-quality topic distributions of short texts by utilizing contrastive learning to model the relations among samples. This work is different from identifying topic boundaries within the document, as done by our work and the related works we discussed and compared to in the paper. Nevertheless, we speculate that encoding the entire topic segment into a semantic space and obtaining a topic-level contrastive representation may help detecting topic boundaries, which we plan to explore in future work.
>
>
> **Q3: Table 2,3,4 are too small to read comfortably**
>
> **Response:** We will adjust the sizes of the tables so that they could be better presented.
>
>
> **Reproducibility: 3**
>
> **Response:** To ensure the transparency and reproducibility of our experiments, we have already provided all the relevant code and experimental results in the supplementary materials. All the training/evaluation datasets are publicly available and references of all datasets are provided in our paper. The code we provided can reproduce the experimental results. We have also provided implementation details in Section 4.1 Experimental Setup and details of hyperparameters in Appendix B (Line 490-491).
>
> [1] Gennady Shtekh, Polina Kazakova, Nikita Nikitinsky and Nikolay Skachkov. 2018. Exploring influence of topic segmentation on information retrieval quality. In Internet Science: 5th International Conference, INSCI 2018, St. Petersburg, Russia, October 24–26, 2018, Proceedings 5, pages 131–140. Springer.
>
> [2] Yang Liu, Chenguang Zhu, and Michael Zeng. 2022. End-to-end segmentation-based news summarization. In Findings of the Association for Computational Linguistics: ACL 2022, pages 544–554.
>
> [3] Zehe A, Konle L, Dümpelmann L K, et al. Detecting scenes in fiction: A new segmentation task[C]//Proceedings of the 16th Conference of the European Chapter of the Association for Computational Linguistics: Main Volume. 2021: 3167-3177.
>
> [4] Rao A, Xu L, Xiong Y, et al. A local-to-global approach to multi-modal movie scene segmentation[C]//Proceedings of the IEEE/CVF Conference on Computer Vision and Pattern Recognition. 2020: 10146-10155.
>
> [5] Luo, Zheheng, Lei Liu, Qianqian Xie and Sophia Ananiadou. “Graph Contrastive Topic Model.” ArXiv abs/2307.02078 (2023): n. pag.
>
> [6] Wu, Xiaobao, Anh Tuan Luu and Xinshuai Dong. “Mitigating Data Sparsity for Short Text Topic Modeling by Topic-Semantic Contrastive Learning.” Conference on Empirical Methods in Natural Language Processing (2022).

---

### Official Review · Reviewer_5wiE · 2023-08-05

**Typos Grammar Style And Presentation Improvements:** 1. Some tables are a bit too small, l…
**Soundness:** 3

**Excitement:**

3: Ambivalent: It has merits (e.g., it reports state-of-the-art results, the idea is nice), but there are key weaknesses (e.g., it describes incremental work), and it can significantly benefit from another round of revision. However, I won't object to accepting it if my co-reviewers champion it.

**Paper Topic And Main Contributions:**

This paper focuses on improving topic segmentation with two auxiliary tasks, topic-aware sentence structure prediction (TSSP) and contrastive semantic similarity learning (CSSL).
TSSP augments a document and then predicts its sentence-pair relations to facilitate sentence representation learning.
CSSL applies contrastive learning to regulate sentence representations by sampling contrastively positive and negative sentence pairs.
This paper conducts extensive experiments to evaluate the performance of the proposed methods.


**Reasons To Accept:**

1. This paper is well-organized and easy to follow.
2. The proposed TSSP and CSSL methods are overall reasonable and easy to understand.
3. The experiments in this paper are extensive.


**Reasons To Reject:**

1. The additional computational cost brought by TSSP and CSSL is unmeasured.

    The cost seems to be high especially because this paper focuses on the topic segmentation of long documents.


2. The way to sample negative samples may be questionable.

    How to obtain negative samples is not reported in detail.
    For example, the blue and yellow sentences in Figure 1 may cover similar topical semantics.
    But CSSL considers them as negative pairs. This will hinder sentence representation learning.
    How to avoid this should be investigated.


3. Some method details are unclear.

    How exactly to replace topics randomly in step 2 of TSSP?

    How exactly to sample negative pairs in CSSL?

**Reproducibility:**

5: Could easily reproduce the results.

**Reviewer Confidence:**

3: Pretty sure, but there's a chance I missed something. Although I have a good feel for this area in general, I did not carefully check the paper's details, e.g., the math, experimental design, or novelty.

---

> ### Author Rebuttal · Authors · 2023-08-29
>
> We thank the reviewer for all valuable feedback. Below we address all your questions and concerns.
>
> **Q1: Additional computational cost brought by TSSP and CSSL is unmeasured**
>
> **Response:** The CSSL module does not introduce any additional parameters. It is used to adjust the sentence representations learned by the encoder and can be computed in parallel to the TSSP module. The TSSP module requires a linear transformation layer with only $768*3$ parameters to predict the structural relationship between sentences. The computational cost of this linear transformation layer is negligible compared to that of the encoder. As mentioned in Section Limitations, the TSSP module works on augmented data, hence it requires twice the training time compared to the baseline model. However, our proposed TSSP and CSSL modules do not change the inference time since we still utilize the same output of encoder and predictor to obtain results. We would like to clarify that Line 537-547 of our paper compares inference speed of the two baselines Longformer-Base and BERT-Base; however, what we should have clarified in the paper is that **our proposed TSSP and CSSL methods do not bring any additional computational cost to inference and do not change inference speed**. We will clarify this point in the revised version.
>
>
> **Q2: The way to sample negative samples may be questionable. The blue and yellow sentences in Figure 1 may cover similar topical semantics. How to avoid this should be investigated.**
>
> **Response:** It is possible that two sentences in different topics could cover similar topical semantics. However, we assume that **two sentences or segments from the same topic are inherently more coherent than those from different topics**. Therefore, the motivation of our CSSL module is to adjust the sentence representation learning to grasp the **relative** high and low coherence relationship. In future work, in order to avoid pulling away sentence pairs in different topics covering similar topical semantics, we plan to consider refining the loss, such as assigning loss weights based on their semantic similarity.
>
>
> **Q3:  Method details are unclear including how to replace topics randomly in TSSP and sample negative pairs in CSSL.**
>
> **Response:** Thank you very much for this question. We apologize for insufficient information on these two details. In Step 2 of Figure 2, for a randomly selected subset of $p_1$ percent of the documents, we replace each topic in them with a topic snippet from other documents with a probability of $p_2$ and keep the same with the probability of $1-p_2$. The default values of $p_1$ and $p_2$ are both 0.5. As for selecting the negative pairs in CSSL, we select $k_2$ sentences from different topics as negative samples, based on the ordering of the distance of a sentence from the anchor sentence, starting from the nearest to the farthest.
>
>
> **Q4: Some tables are a bit too small, like Tables 2 and 3**
>
> **Response:** We will adjust the sizes of Table 2 and 3 so that they are better presented.

---

### Official Review · Reviewer_b6Np · 2023-08-16

**Typos Grammar Style And Presentation Improvements:** The text flows smoothly, with very fe…
**Soundness:** 2

**Excitement:**

3: Ambivalent: It has merits (e.g., it reports state-of-the-art results, the idea is nice), but there are key weaknesses (e.g., it describes incremental work), and it can significantly benefit from another round of revision. However, I won't object to accepting it if my co-reviewers champion it.

**Paper Topic And Main Contributions:**

This paper focuses on the topic segmentation of long documents, a critical task for structuring documents and improving information retrieval. The authors argue that existing supervised neural models for this task have largely overlooked the intricate relationship between semantic coherence and topic segmentation. To address this, the paper introduces two methods: Topic-aware Sentence Structure Prediction (TSSP) and Contrastive Semantic Similarity Learning (CSSL). TSSP is designed to enable the model to understand the structural relationships between sentences in a document, while CSSL aims to ensure that sentence representations within the same topic are semantically similar and distinct from those in different topics. The experimental results show improvements over previous state-of-the-art methods.

**Reasons To Accept:**

- The introduction of TSSP and CSSL offers fresh perspectives and methods to improve topic segmentation.
- Experiments on multiple datasets are provided to demonstrate the gain of the proposed method.
- This paper generally is well-structured and easy to follow.

**Reasons To Reject:**

- The motivation is not solid. While sentence-level coherence is demonstrated to be effective in the experiments, the authors lack a detailed explanation of the underlying mechanism of this method.
- The novelty is limited. I understand that data augmentation and contrastive learning are new to the task of topic segmentation. However, the methods used by the authors do not have a substantial difference from conventional approaches.
- The proposed method is inconsistent with the paper's description. The authors repeatedly mention 'long documents' at the beginning, but in the methodology section, I do not see any design that is specifically tailored for long documents.
- The paper lacks strong and recent baselines for comparison. In the experimental section, the authors compare their approach with multiple baseline models, but these models are either designed for more complex scenarios or have not been published. Such comparisons do not seem to be fair.

**Reproducibility:**

3: Could reproduce the results with some difficulty. The settings of parameters are underspecified or subjectively determined; the training/evaluation data are not widely available.

**Reviewer Confidence:**

3: Pretty sure, but there's a chance I missed something. Although I have a good feel for this area in general, I did not carefully check the paper's details, e.g., the math, experimental design, or novelty.

---

> ### Author Rebuttal · Authors · 2023-08-29
>
> We thank the reviewer for the thoughtful reviews and valuable feedback.  Below are our responses to all of your questions and concerns.
>
> **Q1: The motivation is not solid. Detailed explanation of the underlying mechanism of this method is needed.**
>
> **Response:** The motivation of our work stems from the findings in prior research works that **coherence plays a key role in understanding both logical structures and text semantics**. As summarized in Line 174-200 of our paper, there have been prior works that develop models for comprehending text coherence and tasks to measure effectiveness of the models, such as predicting the coherence score of documents [1], predicting the position where the removed sentence was originally located [2] and restoring out-of-order sentences [3]. Prior works explored incorporating coherence modeling into topic segmentation and achieved better topic segmentation performance. However, **all of prior works consider coherence modeling for topic segmentation only from a single perspective**. [4] only models coherence for understanding only logical structures and forces the model to differentiate coherent sequences of sentences from corrupt ones. [5] models coherence for understanding only text semantics and reduces the semantic coherence scores of the two sentences across topic boundaries.
>
> In contrast to these works, **our work is the first to consider topical coherence as both text semantic similarity and logical structure (flow) of sentences**. As described in Line 092-112 of our paper, we design the two auxiliary coherence-related tasks, Topic-aware Sentence Structure Prediction (TSSP) and Contrastive Semantic Similarity Learning (CSSL), in order to regulate the topic segmentation model to grasp text coherence from both perspectives of logical structures with the TSSP task and text semantic similarity with CSSL. We conducted extensive ablation study to analyze the effect of each module. The ablation results clearly demonstrate that both TSSP and CSSL contribute to the improvements of topic segmentation performance.  Importantly, **these two approaches exhibit complementary effects when combined**, confirming the necessity of modeling both sentence structure and text semantic similarity for identifying topic boundaries.
>
>
> **Q2: Method do not have a substantial difference from conventional approaches**
>
> **Response:**   While it is acknowledged that data augmentation and contrastive learning are well-established techniques in NLP, they have been explored quite insufficiently for topic segmentation. Also, we carefully customized data segmentation and contrastive learning to specifically address the requirements of the topic segmentation task. As discussed in Line 184-200 of our paper, all prior researches exploring coherence for topic segmentation inadequately model the intricate relationship between text coherence and topic segmentation.
> In contrast, **our work is the first to consider both perspectives of text coherence for topic segmentation**, that is, modeling logical structures with our proposed TSSP task and modeling text semantic similarity with our proposed CSSL task. As described in Line 271-291 of our paper, our data augmentation is substantially different from prior data augmentation approaches, as we customize data augmentation for topic segmentation by **simultaneously perturbing the document at both the topic and sentence levels**. We further introduce the TSSP task based on these augmented data to enhance the understanding of logical structures. Also, as described in Line 317-326 of our paper, our proposed CSSL task is also substantially different from conventional contrastive learning approaches. **We customizes constrative learning in CSSL to takes into account both *similar* and *dissimilar* sentence pairs to regulate sentence representations for topic segmentation. Specifically, we design constrative learning in CSSL to promote higher semantic similarity between sentences belonging to the same topic and lower semantic similarity between sentences across different topics**.  The integration of our TSSP and CSSL tasks results in substantial improvements over prior approaches for topic segmentation.
>
>
> **Q3: Any design that is specifically tailored for long documents?**
>
> **Response:** Our work indeed focuses on topic segmentation on long documents due to the reason that all the evaluation datasets used in our work are consisted of long documents. As shown in Section 4.1 of our paper, the datasets used in this paper include In-domain Datasets and Out-of-domain Datasets. As described in Line 386-390, the In-domain Datasets WIKI-727K and English WikiSection datasets are widely used as benchmarks to evaluate the text segmentation performance of models. As described in Line 413-421, we followed prior works and adopted WIKI-50 and Elements datasets as Out-of-domain Datasets. As shown in Table 1, the average number of tokens per document for each of these four evaluation datasets ranges from 1321 to 1654, which exceeds the maximum sequence length 512 of the classic BERT-based pre-trained models. Hence, these datasets are considered long-document datasets.
>
> We specially tailor the backbone model selection for long documents, as elaborated in Line 441-466. We model topic segmentation as a sequence labeling task since prior work [6] shows that sequence labeling is more effective and efficient than paired sentence or chunk modeling for topic segmentation. On this basis, as shown in Table 2, we compare the topic segmentation performance using the BERT-base model with sliding window and several competitive efficient transformers which are specially designed for long sequence modeling, including BigBird-base and Longformer-base, on the WikiSection dataset. As discussed in Line 456-466 and Line 499-502, Longformer-Base outperforms all baselines in the first group of Table 2 and Table 3 and outperforms BERT-Base and BigBird-Base in Table 2. As shown in Table 2, Longformer-Base achieves **82.19 and 72.29 F1**, greatly outperforming BERT-Base (78.99 and 67.34 F1) by **(+3.2, +4.95) F1** and BigBird-Base (80.49 and 70.61 F1). **Hence we select Longformer-Base as the encoder for the main experiments.**  We also show in Figure 3 that the topic segmentation F1 gradually improves as the context length increases, demonstrating the importance of context lengths.
>
> Consequently, on the long-document evaluation datasets, with backbone selected as Longformer to effectively model long documents, we apply our proposed TSSP and CSSL to Longformer, which significantly improves Longformer (and BigBird and BERT) and leads to the SOTA results (Line 510-520).
>
> **It is also important to note that our proposed TSSP and CSSL approaches are agnostic to document lengths and are applicable to models and datasets for short documents**. In future work, we plan to evaluate effectiveness of our methods on short documents or dialogue datasets.
>
>
> **Q4: Lack strong and recent baselines for fair comparison**
>
> **Response:**
> We respectfully disagree on the reviewer's comment that our paper lacks strong and recent baselines for comparison. Firstly, we would like to emphasize that all the evaluation datasets in our experiments are widely used in prior works on text segmentation (the first paragraph in Response to Q3 includes more details). Secondly, **all the baselines we chose for comparison are well-established and exhibit top performance on these benchmarks**. We cited results from all strong and recent baselines and reproduced the best performing prior works that open-sourced their code.
> As noted in Line 467-470, in Table 2, we reproduced results of Longformer-Base on WikiSection with the auxiliary CATS or CSC task described in Section 2.2, which is clearly the strongest baseline. In Table 3, we also reproduced the strong baseline with open-sourced code, Seq-BERT-Base and Seq-ELECTRA-Base. In our experiments, to ensure fair comparisons, we adopted the same partitioning and preprocessing of the datasets as used by the baseline models.
>
> In addition, in order to show that our methods does not require additional label information, we also compare to complex baselines that utilize descriptive labels for each topic, such as Transformer$^{2}_{BERT}$ and Naive LongT5-Base-SS in Table 2. Here we apologize for a typo in Table 2: The last row of group 1 should be Naive LongT5-Base-SS instead of Naive LongT5-Base-DS.  Still, our Longformer-Base baseline already outperforms all baselines of BERT-Base and BigBird-Base.
>
> Upon acceptance of the paper, we plan to make public available all the baseline models we build and reproduce, including Longformer-Base with different context lengths and adding auxiliary coherence tasks from prior works, as well as other backbone models including BERT-Base and BigBird-Base.
>
>
> **Reproducibility: 3**
>
> **Response:** To ensure the transparency and reproducibility of our experiments, we have already provided all the relevant code and experimental results in the supplementary materials. All the training/evaluation datasets are publicly available and references of all datasets are provided in our paper. The code we provided can reproduce the experimental results. We have also provided implementation details in Section 4.1 Experimental Setup and details of hyperparameters in Appendix B (Line 490-491).
>
>
> [1] Regina Barzilay and Mirella Lapata. 2008. Modeling local coherence: An entity-based approach. Computational Linguistics, 34(1):1–34.
>
> [2] Micha Elsner and Eugene Charniak. 2011. Extending the entity grid with entity-specific features. In Proceedings of the 49th Annual Meeting of the Association for Computational Linguistics: Human Language Technologies, pages 125–129.
>
> [3] Somnath Basu Roy Chowdhury, Faeze Brahman, and Snigdha Chaturvedi. 2021. Is everything in order? a simple way to order sentences. In Proceedings of the 2021 Conference on Empirical Methods in Natural Language Processing, pages 10769–10779.
>
> [4] Swapna Somasundaran et al. 2020. Two-level transformer and auxiliary coherence modeling for improved text segmentation. In Proceedings of the AAAI Conference on Artificial Intelligence, volume 34, pages 7797–7804.
>
> [5] Linzi Xing, Brad Hackinen, Giuseppe Carenini, and Francesco Trebbi. 2020. Improving context mod- eling in neural topic segmentation. In Proceedings of the 1st Conference of the Asia-Pacific Chapter of the Association for Computational Linguistics and the 10th International Joint Conference on Natural Language Processing, pages 626–636.
>
> [6] Qinglin Zhang, Qian Chen, Yali Li, Jiaqing Liu, and Wen Wang. 2021. Sequence model with self-adaptive sliding window for efficient spoken document segmentation. In 2021 IEEE Automatic Speech Recognition and Understanding Workshop (ASRU), pages 411–418. IEEE.

---

### Meta-Review · Area_Chair_TaVw · 2023-09-18

**Recommendation:** 3

**Metareview:**

This paper introduces Topic-aware Sentence Structure Prediction (TSSP) and Contrastive Semantic Similarity Learning (CSSL) into modeling long document topic segmentation. The experimental results show their model based on longformer outperforms the existing SOTA methods.

All three reviewers believe that the paper has a good structure and clear organization, and the proposed solution has a certain novelty. However, there are certain concerns about motivation, increasing the cost and impact of using TSSP and CSSL, and the details still need to be further improved. I suggest the author refine the paper based on the valuable feedback from the reviewer, whether accepted or not.

---

### Decision · Program_Chairs · 2023-10-07

**Decision:**

Accept-Main

**Comment:**

This paper introduces Topic-aware Sentence Structure Prediction (TSSP) and Contrastive Semantic Similarity Learning (CSSL) into modeling long document topic segmentation. The experimental results show their model based on longformer outperforms the existing SOTA methods.

All three reviewers believe that the paper has a good structure and clear organization, and the proposed solution has a certain novelty. However, there are certain concerns about motivation, increasing the cost and impact of using TSSP and CSSL, and the details still need to be further improved. I suggest the author refine the paper based on the valuable feedback from the reviewer, whether accepted or not.